# Targeting Striatal Glutamate and Phosphodiesterases to Control L-DOPA-Induced Dyskinesia

**DOI:** 10.3390/cells12232754

**Published:** 2023-11-30

**Authors:** Brik A. Kochoian, Cassandra Bure, Stella M. Papa

**Affiliations:** 1Emory National Primate Research Center, Emory University, Atlanta, GA 30329, USA; brik.kochoian@emory.edu (B.A.K.); cassandra.alyse.bure@emory.edu (C.B.); 2Department of Neurology, Emory University School of Medicine, Atlanta, GA 30329, USA

**Keywords:** L-DOPA-induced dyskinesia, Parkinson’s disease, glutamate, phosphodiesterase, striatal projection neuron, NMDAR, PDE10A

## Abstract

A large body of work during the past several decades has been focused on therapeutic strategies to control L-DOPA-induced dyskinesias (LIDs), common motor complications of long-term L-DOPA therapy in Parkinson’s disease (PD). Yet, LIDs remain a clinical challenge for the management of patients with advanced disease. Glutamatergic dysregulation of striatal projection neurons (SPNs) appears to be a key contributor to altered motor responses to L-DOPA. Targeting striatal hyperactivity at the glutamatergic neurotransmission level led to significant preclinical and clinical trials of a variety of antiglutamatergic agents. In fact, the only FDA-approved treatment for LIDs is amantadine, a drug with NMDAR antagonistic actions. Still, novel agents with improved pharmacological profiles are needed for LID therapy. Recently other therapeutic targets to reduce dysregulated SPN activity at the signal transduction level have emerged. In particular, mechanisms regulating the levels of cyclic nucleotides play a major role in the transduction of dopamine signals in SPNs. The phosphodiesterases (PDEs), a large family of enzymes that degrade cyclic nucleotides in a specific manner, are of special interest. We will review the research for antiglutamatergic and PDE inhibition strategies in view of the future development of novel LID therapies.

## 1. Introduction

Dopamine (DA) replacement therapy with L-DOPA in Parkinson’s disease (PD) has remained the most effective antiparkinsonian treatment since its inception more than half a century ago. However, long-term L-DOPA therapy leads to the development of motor complications that significantly reduce its efficacy. In particular, L-DOPA-induced dyskinesias (LIDs), which are involuntary movements, can affect patients after a few years of therapy. The development of LIDs largely depends on (1) prolonged non-physiologic, pulsatile treatment with L-DOPA or DA agonists and (2) progressive loss of the dopaminergic nigrostriatal system [1]. Other factors, such as abnormal DA release from serotonergic terminals and neuroinflammation, have been implicated in the development of LIDs [2,3]. Although the pathophysiology of LIDs is still not fully understood, it is well established that long-term therapy and progressive disease cause plasticity changes in striatal circuits, namely maladaptive plasticity, that underlie abnormal motor responses to dopaminergic stimulation [4,5,6]. In addition, chronic DA loss is associated with excitability changes in striatal projection neurons (SPNs) that suggest dysregulation of non-dopamine signaling mechanisms [7,8,9].

Several non-dopaminergic mechanisms may be at play to cause the maladaptive plasticity underlying altered SPN responses to DA. In particular, glutamatergic excitation mediated by corticostriatal and thalamostriatal afferents is a major contributor to the SPN plasticity changes. A large body of work supports that in advanced stages of PD, dysregulated glutamate signals lead to hyperexcitability and hyperactivity of SPNs [9,10]. In such a basal excitability state, DA replacement does not restore physiologic SPN outputs to basal ganglia circuits. Furthermore, DA action in hyperactive SPNs induces functional responses (changes in activity patterns) that are typically “unstable” and favor the appearance of LIDs [11,12]. Notably, reducing SPN hyperactivity by administration of NMDA receptor inhibitors stabilizes DA modulation of SPNs and improves LIDs [13,14,15].

DA signaling is mediated by G-protein-coupled receptors that are distributed in two subpopulations of SPNs, i.e., direct pathway SPNs (dSPNs) expressing DA type 1 receptors (D1Rs) and projecting to the internal pallidum and substantia nigra pars reticulata and indirect pathway SPNs (iSPNs) expressing DA type 2 receptors (D2Rs) and projecting to the external globus pallidum [16]. DA receptor activation regulates the synthesis of cyclic adenosine monophosphate (cAMP) and cyclic guanosine monophosphate (cGMP) (Figure 1). These cyclic nucleotides are second messengers that mediate SPN responses to dopaminergic modulation. DA exerts opposite effects on DA receptor subtypes, excitation on D1Rs and inhibition on D2Rs [17]. Therefore, cyclic nucleotides mediate opposite signals in D1R-expressing dSPNs and D2R-expressing iSPNs. However, measurements of global striatal levels of cAMP and cGMP show a decrease in rodent models of LIDs [18], which is difficult to explain by loss of DA modulation of cyclic nucleotide synthesis that is mediated differentially in dSPNs and iSPNs. cAMP and cGMP are catabolized by phosphodiesterases (PDEs) [19], and these enzymes may contribute to changes in cAMP and cGMP levels. Therefore, PDEs may play a role in molecular cascades associated with cyclic nucleotides, the messengers of DA signaling. Indeed, targeting striatal PDE activity with selective inhibitors may have therapeutic potential, particularly restoring physiological SPN responses to DA and thereby reducing LIDs. Several PDE inhibitors have shown anti-LID effects in animal models [20,21], but further studies are necessary to identify specific mechanisms and novel therapeutic targets. Therefore, this review is focused on key LID mechanisms based on maladaptive changes known to contribute to altered DA responses of SPNs, i.e., glutamatergic signals and cyclic nucleotide transduction pathways. 

## 2. Dysregulation of Striatal Projection Neurons

### 2.1. Glutamate Signaling

Both dSPNs and iSPNs are affected by loss of dopaminergic innervation to the striatum in PD [22,23,24,25]. DA depletion leads to significant changes in SPN activity that include upregulated firing and excitability and plasticity changes [9,10,26,27,28]. Ex vivo studies using rodent models showed increased neuronal excitability in the striatum that is positively correlated with DA denervation [7]. Extracellular recordings in advanced parkinsonian non-human primates (NHPs) have shown that spontaneous SPN firing in the basal parkinsonian state is increased by more than 10-fold compared to normal [11,12]. This has also been shown in rat models [25,29,30,31]. Although these recordings in NHPs do not use high-cell-resolution methods, and possibly only iSPNs are hyperactive, the analysis of neuronal responses to L-DOPA suggests that dSPNs are also hyperactive. Similar activity patterns have been observed in patients with PD undergoing deep brain stimulation surgery [32], but due to some conflicting studies [33], the human data may require further analysis. Nevertheless, the available experimental data are highly compelling for upregulated activity of SPNs after DA depletion. This basal hyperactivity may contribute to abnormal L-DOPA responses in neurons that are highly sensitized following chronic loss of DA modulatory inputs [10]. Recordings in parkinsonian NHPs with reproducible LIDs have shown that SPNs exhibit “unstable” responses to DA replacement therapy [11]. The SPN activity changes developed at the onset of the “on” state (reversal of motor disability following DA replacement) are reversed at the peak of L-DOPA effect during the occurrence of dyskinesias [11]. Notably, unstable responses to DA are predominantly found in SPNs with significantly increased basal activity. Thus, the upregulated activity of SPNs underlying altered SPN responses to DA seems to play a role in LID mechanisms. 

Alterations in the strength of excitatory synapses play a pivotal role in the acquisition of motor skills [34]. Synaptic plasticity mechanisms, including long-term potentiation (LTP), long-term depression (LTD), and synaptic depotentiation (SD), require the coordinated activation of multiple inputs onto SPNs [35]. Changes in corticostriatal plasticity have been found in models of PD [6,36], indicating a loss of bidirectional plasticity (LTP and LTD) in both d- and iSPNs, which exhibit only one type (unidirectional) [37]. Notably, in adult parkinsonian mice with stable dopaminergic lesions, iSPNs display LTP and dSPNs display LTD unidirectional corticostriatal plasticity [38]. However, in rodents chronically treated with L-DOPA and exhibiting LIDs, the unidirectional plasticity is reversed, i.e., iSPNs show only LTD and dSPNs show only LTP. Other studies in dyskinetic rats without SPN subtype differentiation showed loss of SD likely to be displayed by dSPNs according to pharmacological tests [39]. Using retrograde cell labeling, Belujon et al. 2010 showed that dyskinesias are correlated with an inability to de-depress established LTD in dSPNs [40]. Therefore, changes in excitability are intricately related to alterations in synaptic plasticity in both subtypes of SPNs. LTP, which is the unidirectional plasticity present in dSPNs in LID models, and the loss of SD in these neurons result in the strengthening of excitatory synapses, further heightening the excitability of these neurons in response to L-DOPA.

Glutamate plays a crucial role in the striatum by driving SPN firing and synaptic plasticity mechanisms. Glutamate signaling is mediated by ionotropic (NMDA, AMPA, and kainate) and metabotropic receptors (mGluRs). Ionotropic receptors are ligand-gated ion channels that produce a fast depolarization of the cell membrane and are implicated in various aspects of synaptic plasticity [35]. Conversely, mGluRs mediate signal transduction through phosphoinositide (PI) and cyclic nucleotide messenger systems that may produce slower and more prolonged excitatory or inhibitory responses. mGluRs are subdivided into three groups based on G-protein coupling and ligand profiles [41]. Group I receptors (mGluR1 and mGluR5) mediate excitatory responses, while groups II (mGluR2 and mGluR3) and III (mGluR4, mGluR6, mGluR7, and mGluR8) are inhibitory [41]. Group I receptors are postsynaptic on SPNs, while groups II and III are presynaptically located on corticostriatal terminals and GABAergic fibers [41]. The extended distribution of glutamate receptors with different functions emphasizes their importance in striatal circuit mechanisms. 

In PD, DA loss significantly impacts striatal glutamatergic transmission, which can be interpreted as a compensatory mechanism in the SPN microcircuitry [41]. Importantly, these compensatory changes in glutamatergic activity within the striatum occur independently of DA or GABA levels [41], suggesting that glutamatergic function is primarily dysregulated [41]. In addition, chronic L-DOPA therapy has been shown to alter glutamate receptor expression [41], which may facilitate signal transmission. Altogether, data support that glutamatergic hyperactivity is associated with altered SPN responses to DA [14]. These changes underlie the emergence of involuntary movements in response to L-DOPA [41]. Table 1 summarizes the data relevant to the role of the glutamate system in LIDs.

### 2.2. Dopamine Signaling

DAR signaling is predominantly mediated through the cAMP/protein kinase A (PKA) molecular cascade [63] (Figure 1). D1R activation stimulates Gs/olf proteins, activating adenylyl cyclase (AC) to synthesize cAMP in dSPNs [64,65]. Conversely, D2R activation stimulates Gi/o proteins, inhibiting AC activity and thereby reducing cAMP levels in iSPNs. Notably, adenosine 2A receptors (A2AR) in iSPNs counteract D2Rs by activating AC, increasing cAMP synthesis, likely for preventing excessively low levels of cAMP. A2ARs and D2Rs form heteromers in the striatum, which leads to macromolecular complexes with different functionality than their individual receptors [66,67]. Of importance, recent work has further described the high proportion of receptors that display this interaction, emphasizing the development of A2AR-based antiparkinsonian therapy [68]. cAMP serves as a key modulator of various physiological processes that include PKA (regulations of Ca^2+^ levels and the function of various proteins) and cyclic nucleotide-gated channel (CNGC) mechanisms [69,70]. Activated PKA can also phosphorylate transcription factors, leading to gene expression changes affecting ion channels, synaptic plasticity, and neuronal excitability [69,70].

cGMP synthesis can be triggered through the activation of either membrane-bound particulate guanylyl cyclase (pGC) or soluble guanylyl cyclase (sGC) by natriuretic peptides and nitric oxide (NO), respectively [71,72]. NO is a gaseous free radical produced by GABAergic interneurons co-expressing neuronal NO synthase (nNOS), neuropeptide Y, and somatostatin [73] and requires concurrent NMDA receptor and D1R activation [74]. Upon binding to sGC, NO enhances GC activity, converting guanosine 5′-triphosphate (GTP) to cGMP. cGMP activates the cGMP/PKG molecular cascade and cGMP-gated ion channels [63]. Both cAMP and cGMP are crucial second messengers of DA signaling in SPNs.

### 2.3. Modulation of Dopamine Signaling by PDEs

Cyclic nucleotides are critically involved not only in dopaminergic but also glutamatergic signal transduction mechanisms. Drugs that increase cyclic nucleotide levels or reduce their degradation have been shown to facilitate corticostriatal transmission, while those that decrease cyclic nucleotide levels or increase their degradation can suppress it [75]. Interestingly, reductions in cGMP, but not cAMP, have been reported in animal models and patients with PD [76,77]. However, both striatal cAMP and cGMP levels are significantly lower at the peak of LID in rodent models [18,78]. cAMP and cGMP levels are regulated by PDEs, which are enzymes that hydrolyze the cyclic bond converting them into inactive 5′-AMP and 5′-GMP, respectively. Different isoforms of PDEs are expressed in distinct brain regions and cell types, allowing for precise modulation of cyclic nucleotide-mediated DA signaling. Numerous studies have demonstrated that PDE inhibitors can elevate intracellular nucleotide levels and facilitate membrane excitability and responsiveness of SPNs to corticostriatal inputs [75,79,80]. The function of PDEs in various neurological disorders, including PD, has recently received significant attention, even though the specific roles of PDEs in the pathophysiology of PD symptoms, including LIDs, remain to be investigated.

## 3. Targeting Glutamatergic Dysfunction

### 3.1. Ionotropic Glutamate Receptors

Following DA loss and chronic DA replacement therapy, various molecular, structural, and functional alterations occur in ionotropic glutamate receptors, including changes in their expression, composition, trafficking, and localization [81,82,83,84]. Striatal overexpression of NMDA and AMPA receptors has been shown in animal models and patients with PD [44,85]. Notably, there is an increased ratio of NMDA receptors (NMDARs) to AMPA receptors (AMPARs) following DA depletion as well as a higher ratio of GluN2A to GluN2B subunits after prolonged exposure to L-DOPA and the development of LIDs [42,46,47,48]. These changes may amplify excitatory glutamatergic transmission at corticostriatal synapses, a mechanism strongly implicated in LID pathophysiology [86,87,88,89,90].

Additionally, elevated levels of glutamate release and NMDAR activation have been observed in dyskinetic PD patients following L-DOPA administration [50]. Intrastriatal microinjection of AMPAR or NMDAR antagonists in parkinsonian and dyskinetic NHPs reduces basal SPN hyperactivity and counteracts the typical “unstable” response to L-DOPA, restoring stable SPN responses of non-dyskinetic parkinsonian NHPs [14]. Significantly, in these studies, the application of NMDAR antagonists across an extended area of the putamen in parkinsonian NHPs significantly reduces LIDs. Studies that tested the systemic administration of AMPA antagonists showed contradictory results likely because of different experimental paradigms [91,92]. Of interest, glycine (a co-agonist on NMDARs) antagonists improved the antiparkinsonian action of L-DOPA in a study of hemiparkinsonian NHPs and LIDs in parkinsonian rats and NHPs [93,94,95,96]. The systemic administration of selective NMDAR ion channel blockers and competitive antagonists significantly reduces dyskinesias [13,51,97]. These early tests used agents not viable for further development due to toxicity, but results served as proof of concept, and, thus, several non-competitive NMDAR antagonists were subsequently tested in rodent and primate models. Examples include amantadine, dextromethorphan, and memantine [98,99,100,101,102]. However, the pharmacological profiles of these drugs whose properties included partial antagonistic action, lower affinity, and multiple binding sites weakened their specific efficacy for controlling LIDs. In addition, a common problem with targeting NMDARs is their wide distribution in the brain, which favors dose-dependent off-target effects of selective agents [103,104]. One strategy to overcome this problem has been to develop subunit-selective inhibitors based on the predominant subunit composition of NMDARs in the striatum. Studies conducted with an NR2B-selective NMDAR antagonist, CP-101,606 (Pfizer, Inc., New York, NY, USA), have yielded inconsistent results in animals models, improving dyskinesia and augmenting the antiparkinsonian action of L-DOPA in some models and exacerbating LIDs in others [105,106]. Therefore, the experimental work provided evidence to support that therapeutic interventions targeting dysregulated glutamatergic signaling can reverse altered SPN responses to L-DOPA and control LIDs. Table 2 summarizes the relevant studies of agents targeting glutamatergic transmission.

### 3.2. Metabotropic Glutamate Receptors

The role of mGlu receptors (mGluRs) in LID pathophysiology has been extensively studied in search of more moderate effects that could have less toxicity. Within the mGluR family, mGluR5, a group I receptor, has garnered the most focus, whereas the role of mGluR1 remains less well defined, partly due to its lower presence in the striatum than mGluR5 [107,108]. Studies indicate that mGluR5 levels in the striatum correlate with LID development and are elevated in animal models and patients with LIDs [41,52,55]. Moreover, dyskinesia severity is linked to higher mGluR5 levels in parkinsonian NHPs with LIDs [53,54]. Genetic studies in mice that compared NMDAR-evoked responses between animals lacking mGluR1 and mGluR5 have shown a prominent role of mGluR5 in NMDAR signaling [109]. These findings highlight the specific role of mGluR5 in LID pathophysiology and have fueled interest in developing mGluR5 antagonists as therapeutic agents for LIDs.
cells-12-02754-t002_Table 2Table 2Antiglutamatergic pharmacotherapies.ReceptorFindingsSpeciesReferencesNMDARNMDAR antagonists (e.g., amantadine, memantine, CP-101,606, and ramacemide) reduce LIDs (some studies also claim antiparkinsonian effects)RatPapa et al., 1995; Tronci et al., 2014 [51,98]NHPSingh et al., 2018 [14]HumanNutt et al., 2008; Shoulson et al., 2001 [104,110]AMPARAMPAR antagonists reduce LIDsRatKobylecki et al., 2010 [91]NHPKobylecki et al., 2010 [91]

AMPAR antagonists have no antidyskinetic effectNHPHuman
Luquin et al., 1993 [92]Eggert et al., 2010 [111]Group ImGluRsmGluR5 NAMs reduced LIDsRatNHPHumanRylander et al., 2009 [112]Morin et al., 2010 [113]McFarthing et al., 2019 [114]mGluR1 antagonists modestly improved LIDsRatRylander et al., 2009 [112]

Group IImGluRsmGluR2/3 agonist improved LID durationRatZheng et al., 2020 [102]

Group IIImGluRsmGluR4 PAM has antidyskinetic effectsRatNHPCalabrese et al., 2022 [62]Charvin et al., 2018 [115]
mGluR4 PAMs “spare” L-DOPA, reducing dose requirementRatIderberg et al., 2015 [116]


RatHumanLe Poul et al., 2012 [117]mGluR4 PAMs have antiparkinsonian effects but no effect on LIDs

Rascol et al., 2022 [118]


The mGluR5 negative allosteric modulator (NAM) 2-nethyl-6-(phenylethynyl)pyridine (MPEP) prevented the observed increase in mGluR5 expression following L-DOPA treatment in rodents [54,60] and reduced LIDs in parkinsonian NHPs [113]. Other mGluR5 NAMs, such as dipraglurant and mavoglurant, also demonstrated antidyskinetic effects without affecting the antiparkinsonian action of L-DOPA in MPTP-treated NHPs [119,120]. In contrast, mGluR1 antagonists have only modestly improved dyskinesia in animal models [112].

Preclinical studies also explored targeting presynaptic group II mGluRs to control LIDs by inhibiting glutamate release, thereby reducing the excitatory drive on SPNs and subthalamonigral terminals for improving parkinsonism [61,121,122]. However, translating this approach into effective LID treatments using mGluR2/3 agonists has yielded inconsistent results [90,112]. A more recent study compared amantadine to LY354740, an mGluR2/3 agonist, in parkinsonian rats and found that LY354740 weakly reduced LID duration but had no effect on LID intensity and interfered with the antiparkinsonian action of L-DOPA [102]. Consequently, group II agonists are unlikely to be as effective as group I mGluR5 antagonists for LID therapy.

Group III mGluRs, notably mGluR4, are emerging as therapeutic targets for LIDs. mGluR4 is also presynaptically expressed in corticostriatal synapses, reducing glutamate release. In vitro recordings of corticostriatal slices revealed that foliglurax decreases spontaneous glutamatergic transmission and, when co-administered with L-DOPA, restores bidirectional plasticity in the striatum [62]. Foliglurax, a selective positive allosteric modulator (PAM) at mGluR4, has antidyskinetic effects in parkinsonian rats and NHPs [115]. Foliglurax likely also modulates GABA release from iSPNs, which can mediate antiparkinsonian effects and reduce L-DOPA requirements [123]. Thus, some studies of mGluR4 PAMs in parkinsonian models also showed antiparkinsonian effects without a measurable impact on LIDs [116,117,124]. Nonetheless, further exploring mGluR4 PAMs also holds promise for improving LIDs indirectly by reducing L-DOPA dose requirements. Overall, mGluR5 and mGluR4 seem like better targets for LID treatment than mGluR1 and group II mGluRs (Table 2).

### 3.3. Recent Clinical Trials

Early clinical trials of NMDAR antagonists included ramacemide and CP-101,606, which were reported to decrease LIDs and augment the antiparkinsonian effects of L-DOPA [104,110]. Amantadine has been studied in multiple trials, showing a reduction of LID severity without affecting L-DOPA’s efficacy (ClinicalTrials.gov Identifiers: NCT02136914, NCT02274766). The antidyskinetic effect of amantadine is thought to primarily result from its inhibition of NMDARs, reducing glutamatergic hyperactivity [125], although it also has other effects, including preventing the abnormal reductions in cyclic nucleotides linked to LIDs in animal models of PD [78]. Of note, amantadine has also demonstrated anti-inflammatory effects in both in vitro and in vivo models of neuroinflammation and PD, leading to a reduction in levels of TNF-α, IL-1β, and NO [126,127]. Since neuroinflammation is thought to contribute to LIDs, the antidyskinetic effects of amantadine may be partially attributed to a reduction of neuroinflammation. Extended-release amantadine is the sole recently approved medication for the chronic treatment of LIDs, although the instant-release formulation has been used acutely for over 50 years [125]. It is important to note that amantadine can produce significant adverse events, including visual hallucinations, confusion, blurred vision, and gastrointestinal symptoms, which underscore the need for developing alternative treatments [125]. Glycine binds to the NMDAR as a co-agonist, contributing to opening of the ion channel, and the glycine receptor antagonist AV-101 (L-4-chlorokynurenine, VistaGen Therapeutics, Inc., South San Francisco, CA, USA) is in a Phase 2 trial for LID therapy (ClinicalTrials.gov Identifier: NCT04147949).

The mGluR5 NAM dipraglurant (ADX48621, Addex Therapeutics, Geneva, Switzerland) was tested in a Phase 2a trial (ClinicalTrials.gov Identifier: NCT04857359) for reducing peak-dose LIDs and showed positive effects [114]. However, other mGluR5 NAMs had only mild antidyskinetic effects, worsened motor function, and had increased adverse events [128,129,130]. Foliglurax (Prexton Therapeutics, Oss, The Netherlands), the first mGluR4 PAM evaluated clinically, failed to meet Phase 2 trial endpoints (ClinicalTrials.gov Identifier: NCT03162874) and showed no measurable improvement of LIDs [118]. Since the use of non-specific drugs targeting glutamate receptors typically produces off-target effects [104], clinical trials use narrow dose ranges to minimize adverse reactions at the expense of compromising efficacy [103]. Overall, antiglutamatergic pharmacotherapies have shown variable results for controlling LIDs (Table 2), with adverse events listed as the major limitation for their use [104,125].

## 4. Targeting Phosphodiesterases

### 4.1. Families and Properties of PDEs

The PDE superfamily, consisting of 21 genes and 11 subtypes with over 100 splice variants, controls cyclic nucleotide catabolism [19,131]. PDEs are an ideal target to manipulate signal transduction mechanisms in a region-specific manner since isoenzymes have distinct distributions in the brain [132,133]. Furthermore, PDEs exhibit different substrate affinities, selectively catabolizing cAMP, cGMP, or both, which enables precise control over the levels of a specific cyclic nucleotide. This specificity is particularly relevant to DA signaling pathways that may differentially utilize cAMP or cGMP, thereby offering a unique opportunity for developing precise drug therapies to modulate L-DOPA responses (Table 3). PDE isoforms with mRNA expression in the striatum include the dual-substrate enzymes PDE1B, PDE2A, and PDE10A that metabolize cAMP and cGMP as well as the cAMP-specific enzymes PDE3A, PDE3B, PDE4D, PDE7B, and PDE8B, along with the cGMP-specific enzyme PDE9A [133]. In recent years, PDE inhibitors have garnered attention for their potential therapeutic applications in various disorders that are associated with striatal pathophysiology, such as schizophrenia, Huntington’s disease, and PD [76,77,134,135]. We will review the most relevant research concerning PDE isoenzymes as potential targets for LID therapy.

### 4.2. Preclinical Research in Animal Models

#### 4.2.1. Phosphodiesterase 10

PDE10A is expressed almost exclusively in SPNs (with very low mRNA expression found in other brain areas), localizing primarily in the membranes of dendrites and dendritic spines. This enzyme regulates the levels of cAMP and cGMP [133,136,137]. Given the restricted expression of PDE10A to the striatum, targeting PDE10A could have an improved safety profile compared with other PDEs that have a wide distribution in the CNS [69,138].

PDE10A expression and activity are decreased in the striatum and increased in the nucleus accumbens in parkinsonian rats [139]. Accordingly, cAMP levels were upregulated in the striatum and downregulated in the nucleus accumbens. However, cGMP levels were decreased in all DA-depleted regions [139], suggesting that other mechanisms intervene in striatal cGMP regulation in the context of PD [139]. Positron emission tomography (PET) imaging of (11)C-IMA107, a highly selective PDE10A radioligand, revealed lower PDE10A availability in the caudate, putamen, and globus pallidus in patients with PD than in healthy controls [140]. The decrease in PDE10A binding correlated with disease duration motor disability and LIDs [140].

Several studies that examined PDE10A inhibition in models of PD found antidyskinetic effects [20,141,142]. In parkinsonian rats, slight inhibition of PDE10A attenuated abnormal involuntary movements (AIMs; the rodent equivalent of LIDs) without interfering with cortically evoked activity of striatal neurons [142]. More marked inhibition of the enzyme using higher inhibitor doses resulted in AIM prolongation and increased cortically evoked activity [142]. These findings suggest that facilitating corticostriatal transmission through PDE10A inhibition is likely to contribute to AIMs. However, PDE10A inhibition activates iSPNs to a greater extent than dSPNs, suggesting a predominant action of PDE10A inhibition counteracting DA modulation in the striatopallidal pathway [69,143,144]. Therefore, the dose-dependent effect of PDE10A inhibition may be attributed to the selective activation of iSPNs at lower doses and non-selective activation of both iSPNs and dSPNs at higher doses. Given these findings, further analysis of the functional changes mediated by PDE10A inhibition with cell-type resolution is warranted.

Studies in NHPs showed that selective PDE10A inhibitors produce behavioral effects that are akin to D2R antagonism but distinct from typical antagonists, likely due to the combined D1R agonism-like effect [145,146]. A PDE10A inhibitor, MR1916 (Mochida Pharmaceuticals, Inc., Shinjuku City, Japan), reduced dyskinesia without interfering with L-DOPA’s efficacy in parkinsonian rats and NHPs [20]. In the primate study, the dose of MR1916 that produced the maximal antidyskinetic effect was not the highest dose tested, supporting the notion that the antidyskinetic properties of PDE10A inhibition are driven by facilitation of the striatopallidal, rather than striatonigral, pathway. Nevertheless, contradictory results have been reported regarding the role of PDE10A in PD and LIDs, and, thus, further investigation is needed to better understand the mechanisms and therapeutic potential of PDE10A inhibition.
cells-12-02754-t003_Table 3Table 3Properties of PDEs.FamilySubstrateRegulationCNS ExpressionInhibitor (s)Role in PD and LIDsReferencesPDE1cAMP/cGMP
Cortex, hippocampus, cerebellum, striatum, nucleus accumbens, olfactory bulb, amygdala, thalamusVinpocetine, ITI-214, DSR-143136Upregulated in parkinsonian rats; genetic KO enhances responsiveness to DA agonists; inhibited by antiparkinsonian medications; antidyskinetic effect of inhibitors in NHPsSancesario et al., 2004; Lakics et al., 2010; Cenci et al., 2018; Enomoto et al., 2021; Kakkar et al., 1997; Kakkar et al., 1996; Polli et al., 1994; Reed et al., 2002 [77,133,147,148,149,150,151,152]PDE2cAMP/cGMPcGMP stimulatedCortex, hippocampus, striatum, hypothalamus, amygdala, midbrainLu AF64280, BAY 60-7550Upregulated activity in dyskinetic ratsSancesario et al., 2014; Lakics et al., 2010; Doummar et al., 2020; Heckman et al., 2017; Salpietro et al., 2018; Simpson et al., 2010; Stephenson et al., 2009 [78,133,153,154,155,156,157]PDE3cAMPcGMPinhibitedHippocampus, striatumCilostazol
Lakics et al., 2010 [133]PDE4cAMP
Cortex, hippocampus, striatum, olfactory bulb, thalamus, hypothalamus, cerebellum, midbrainRolipram, ND1251, MK-0952, MEM1414, HT-0712, roflumilast, DenbufyllineModulates A2AR signaling; enhances A2AR- and D1R-mediated phosphorylation of DARPP-32; linked to LIDs in animal modelsNishi et al., 2008; Lakics et al., 2010; Casacchia et al., 1983; Cherry et al., 1999; Perez-Torres et al., 2000; Vignola et al., 2004; Yang et al., 2008 [69,133,158,159,160,161,162]PDE5cGMP
Cerebellum, spinal cord, cortex, hippocampusSildenafil, udenafil, vardenafil
Lakics et al., 2010 [133]PDE6cGMP
Retinal rod, cone cells, pineal gland

Lakics et al., 2010 [133]PDE7cAMP
Hippocampus, striatum, cerebellum, cortex, thalamus, hypothalamus, midbrainS14, BRL-50481Upregulated in degenerating DA cells; neuroprotective effects of inhibitorsLakics et al., 2010; Casacchia et al., 1983; Ciccocioppo et al., 2021; Morales-Garcia et al., 2015; Morales-Garcia et al., 2020; Morales-Garcia et al., 2011 [133,158,163,164,165,166]PDE8cAMP
Cortex, hippocampus, striatum, cerebellum, olfactory bulb, thalamus, hypothalamus, midbrainNon-chiral 9-benzyl-2-chloro-adenine derivatives, PF-04957325Potential role in motor performance and coordinationLakics et al., 2010; Golkowski et al., 2016; Kobayashi et al., 2003; Wu et al., 2022 [133,167,168,169]PDE9cGMP
Striatum, cerebellum, thalamus, hypothalamus, amygdala, olfactory bulb, cortex, hippocampus, midbrainPF-04447943, BI 409306, FRM-16606Inhibition enhanced antiparkinsonian action of L-DOPA in NHPs; no direct effect on LIDsMasilamoni et al., 2022; Lakics et al., 2010 [21,133]PDE10cAMP/cGMPcAMP inhibitedStriatum (hippocampus, cortex, midbrain, and cerebellum: very low mRNA levels detected)MP-10, TAK-063, RO5545965, AMG 579, OMS824, PDM-042Inhibition shows antidyskinetic effects in animal models; regulates corticostriatal transmission; altered expression in PD patientsBeck et al., 2018; Addy et al., 2009; Lakics et al., 2010; Xie et al., 2006; Niccolini et al., 2015; Arakawa et al., 2020; Guimaraes et al., 2022; Schmidt et al., 2008; Uthayathas et al., 2014; Lenda et al., 2021 [20,103,133,136,140,141,142,145,146,170]PDE11cAMP/cGMP
Hippocampus

Lakics et al., 2010 [133]


#### 4.2.2. Phosphodiesterase 1

PDE1 is a family of enzymes that can also hydrolyze both cAMP and cGMP and is comprised of three genes: PDE1A, PDE1B, and PDE1C. PDE1B highly co-localizes with D1Rs in the frontal cortex and striatum [133], and, therefore, it is believed to primarily affect the function of dSPNs [151]. Studies of the enzyme levels showed upregulation in parkinsonian rats [77], while genetic knockout in mice enhanced responsiveness to DA agonists and increased locomotor activity [152]. A recent study reported that the PDE1 inhibitor DSR-143136 (Sumitomo Dainippon Pharma Co., Ltd., Osaka, Japan) reversed LIDs in parkinsonian NHPs without adversely affecting parkinsonian motor symptoms [148]. DSR-143136 is highly selective for PDE1B over PDE1A and PDE1C and exhibits brain penetration in rodents, lending credibility to its potential therapeutic application [148,171] and supporting further studies of PDE1 inhibition.

#### 4.2.3. Phosphodiesterase 2

PDE2 is another dual-substrate enzyme, and one of its isoforms, PDE2A, is highly expressed in the striatum [133]. Immunoreactivity studies have shown that PDE2A is more abundant in the axons of SPNs than in their cell bodies [157]. Most research on PDE2 has focused on Alzheimer’s disease and schizophrenia, showing potential benefits for memory and cognitive function [154], which hypothetically could be applicable for the treatment of non-motor symptoms in PD. PDE2A is activated by cGMP binding to its allosteric site located in the N-terminal domain [157,172]. Notably, increased activity of cGMP-specific PDEs has been observed in dyskinetic rats, suggesting that inhibition of this enzymatic activity may be antidyskinetic [78]. Conflicting with this notion, mutations in PDE2A genes that result in reduced enzymatic activity have been associated with several other forms of dyskinesia, including chorea and pleiotropic fluctuation/paroxysmal dyskinesias [153,155]. Thus, the future of targeting PDE2A activity for LID therapy is uncertain. PDE11 is another dual-substrate enzyme, but its expression is predominantly in the hippocampus.

#### 4.2.4. Phosphodiesterase 4

In the group of cAMP-selective PDEs expressed in the striatum, PDE3 has not been studied for motor effects in the context of PD. PDE4 is encoded by four genes: PDE4A, PDE4B, PDE4C, and PDE4D. PDE4B is the most abundant isoform in the CNS and is largely distributed through brain regions [133]. In the striatum, PDE4A, PDE4B, and PDE4D are expressed, but PDE4B is the most abundant [160,173]. PDE4B is expressed in both dSPNs and iSPNs, but immunohistochemical analysis revealed higher expression within the iSPNs [69].

Rolipram, a PDE4 inhibitor, weakly enhances cAMP/PKA signaling in the striatum both in vitro and in vivo [69]. This enhancement is aligned with adenosine A2A receptor (A2AR)-mediated signaling, which counteracts the action of D2Rs. Rolipram treatment increased A2AR-mediated phosphorylation of DARPP-32, a key signaling molecule, but not D1R-mediated phosphorylation of DARPP-32. Therefore, PDE4 inhibition is believed to primarily enhance adenosine signals in the striatal indirect pathway [69]. This effect of PDE4 inhibition counteracts DA activation of D2Rs, an effect similar to D2 receptor antagonism. Rolipram has also been reported to have protective effects for MPTP-induced neurodegeneration in models of PD [162].

#### 4.2.5. Phosphodiesterase 7

PDE7 is another isoenzyme with specific affinity for cAMP, as shown by the effect of selective inhibitors significantly elevating cAMP levels in the CNS [163]. PDE7 has two known isoforms, PDE7A and PDE7B, each with distinct tissue distribution and subcellular localization. While mRNAs of both isoforms are expressed in the brain and peripherally throughout different body tissues, PDE7B predominates in the brain [133,174]. Based on upregulation of PDE7 in degenerating DA cells and microglia in various experimental models, studies of PDE7 inhibition have been focused mainly on neuroprotection [165]. S-14 (a non-selective inhibitor) and BRL50481 (a selective inhibitor) reduced microglial activation and neuronal cell loss [164,166]. These studies also reported improved apomorphine-induced asymmetric motor behavior, suggesting that these agents may potentiate the antiparkinsonian action of DA agonists. However, the rodent model used in this study was produced with lipopolysaccharide, an endotoxin that triggers neuronal damage via neuroinflammation. Therefore, the translatability of these results is unclear, and data have not yet been replicated in other models of PD. PDE8, which also specifically hydrolyzes cAMP and has some expression in the striatum [133,168], has not been investigated in the context of PD.

#### 4.2.6. Phosphodiesterase 9

Three PDE isoenzymes (PDE5, PDE6, and PDE9) have specificity for cGMP, but PDE5 and PDE6 have very low striatal expression [175,176]. PDE9A is widely distributed in the brain and is abundant in the striatum [133]. Therefore, PDE9 inhibition is an experimental tool that can facilitate the study of mechanisms mediated by the cGMP-PKG signaling pathway. Several selective inhibitors have been used in parkinsonian models of PD and LIDs. The PDE9 inhibitor FRM-16606 (Forum Pharmaceuticals, Inc., Waltham, MA, USA) prolongs the antiparkinsonian action of L-DOPA with a slight increase in LIDs [21]. However, it is important to note that the LID increase was likely due to the longer duration of the L-DOPA response rather than a direct effect of the PDE9 inhibitor on LID mechanisms. PDE9 inhibition did not induce dyskinesias when administered without L-DOPA and did not augment LID scores at the peak of the L-DOPA response. The lack of antidyskinetic effects of PDE9 inhibition may be attributed to the different roles of the cAMP-PKA and cGMP-PKG pathways in LID mechanisms. Data also suggest that the NO-cGMP-PKG pathway is an important signaling mechanism in DA-mediated reversal of parkinsonian motor deficits. Striatal cGMP levels, but not cAMP levels, are decreased in parkinsonian animals [18,77], which may explain the synergistic effect of the PDE9 inhibitor with L-DOPA antiparkinsonian action. Of note, it is unclear how the NO-cGMP-PKG signaling pathway is modulated by D1R or D2R activation in the intact and DA-depleted striatum. Thus, PDE9-mediated mechanisms in distinct SPN subtypes and correlated motor effects remain to be investigated.

### 4.3. Recent Clinical Trials

While several PDE4 inhibitors have been investigated in clinical trials for neurological disorders, dose-limiting adverse events have often been determinant for low efficacy [161]. For example, rolipram at the doses tested did not improve the efficacy of L-DOPA or other dopaminergic drugs [158]. The extensive expression of PDE4 in the basal ganglia and its role in modulating adenosine and DA signaling pathways support further research for developing more selective and better-tolerated PDE4 inhibitors as potential therapeutic agents for PD.

Among numerous clinical trials that have explored PDE inhibitors for various cognitive and motor disorders, including antiparkinsonian effects [158], only one clinical trial was designed to determine the effects on LIDs. This ongoing trial of a selective PDE10A inhibitor, CPL500036 (Celon Pharma SA), is in Phase 2 (ClinicalTrials.gov Identifier: NCT05297201). Contrary to most experimental data on PDE10A inhibitors, the preclinical study that had investigated the efficacy of CPL500036 in hemiparkinsonian rats found antiparkinsonian-like effects, reversing impairments in contralateral forelimb function [170]. Notably, this study did not investigate CPL500036′s impact on LIDs and reported sedation in parkinsonian rats at therapeutic doses [170].

## 5. Concluding Remarks

Striatal maladaptive plasticity is highly involved in SPN responses to L-DOPA underlying dyskinesias. A large body of work has shown that glutamatergic dysregulation is a major contributor to the reduced homeostatic modulation of SPNs by DA replacement [81,82,83,84]. Ionotropic glutamate receptors undergo significant changes following DA loss and chronic replacement therapy, including alterations in expression, composition, and subunit ratios [44,47,48,85]. Such changes impact plasticity mechanisms at corticostriatal synapses. Metabotropic glutamate receptors also play a role, either modulating postsynaptic signaling pathways or regulating glutamate release [55]. On the other hand, recent data have also highlighted the role of altered DA signaling transduction in LID mechanisms. Changes in DA signal transduction pathways occur after chronic L-DOPA treatment, as indicated by the levels of cyclic nucleotides. Striatal cAMP and cGMP levels are lower in models of LIDs [18,78]. Therefore, there is a growing interest in the regulation of cyclic nucleotides, which is mediated by the catabolic PDE enzymes.

The review of preclinical and clinical work on glutamatergic agents lead to the conclusion that selective NMDAR antagonists and mGluR agents have antidyskinetic effects, but efficacy has been limited by the target characteristics (e.g., receptor distribution) and pharmacological profiles of available drugs [98,99,100,101,102,125]. Strategies to circumvent this relying on identifying receptor properties that could confer more region specificity are yet to be proven successful. Some examples would be activity-dependent binding agents or particular receptor biophysical features (binding kinetics and pH changes) [177,178,179]. Gene therapy provides an alternative approach for specific manipulation of striatal glutamate signaling. One study in rodents that assessed selective mGluR5 knockdown in dSPNs showed positive results [180]. Overall, there is compelling evidence for the link between striatal glutamate and LID mechanisms and to support further studies of this target to develop effective antidyskinetic therapies.

In turn, PDE inhibitors, including those selective for isoforms with dual- or single-substrate affinity, are of interest for LID treatment. PDE10A and PDE1B inhibition (dual action) have been studied in parkinsonian rats and NHPs, providing proof of concept that targeting cyclic nucleotide regulation may produce antidyskinetic effects [20,148]. PDE9 is the only isoenzyme that has specificity for cGMP (single action) and significant expression in the striatum. Selective PDE9 inhibition enhances the antiparkinsonian action of L-DOPA with little to no impact on LIDs [21]. Also important, PDEs with single-cAMP action have not yet been explored in the context of PD/LIDs. Therefore, the available data warrant further studies on DA signal transduction mechanisms and the role of different PDE targets in responses to DA replacement. Research into PDE inhibition is emerging as a new therapeutic avenue to control LIDs and improve the quality of life of patients with PD.

## Figures and Tables

**Figure 1 cells-12-02754-f001:**
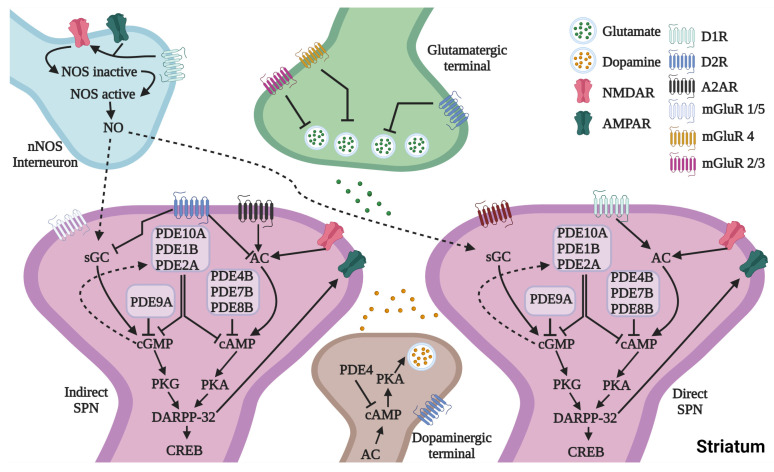
Microcircuitry of striatal projection neurons. The scheme is an oversimplification of the microcircuitry. DA stimulation modulates the activity of the d- and iSPNs. DAR signaling is mediated by activation of cyclic nucleotide synthesis through two pathways (AC-cAMP and NO-GC-cGMP). PDEs are catabolic enzymes for cAMP and cGMP (PDE10, PDE1, PDE2, PDE4, PDE7, and PDE9 are abundant in the striatum). SPNs receive excitatory signals mediated by glutamate ionotropic (NMDAR and AMPAR) and metabotropic receptors (groups I–III), the latter also acting presynaptically to modulate glutamate release. Arrows represent activation, and perpendicular lines represent inactivation. Abbreviations: SPN = striatal projection neuron; PDE = phosphodiesterase; NOS = nitric oxide synthase; NO = nitric oxide; GC = guanylate cyclase; cGMP = cyclic guanosine monophosphate; PKG = protein kinase G; AC = adenylate cyclase; cAMP = cyclic adenosine monophosphate; PKA = protein kinase A; DARPP-32 = dopamine- and cAMP-regulated neuronal phosphoprotein; CREB = cAMP response element binding protein.

**Table 1 cells-12-02754-t001:** Glutamate transmission mechanisms.

Receptor	Findings	Species	References
NMDAR	Increased ratio of NMDARs/AMPARs	Rat	Bagetta et al., 2012 [42]


Overexpression of receptors in the striatum	RatNHPHuman	Chase et al., 2000 [43]
Calon et al., 2002 [44]
Calon et al., 2003 [45]
Higher ratio of GluN2A/GluN2B subunits after prolonged exposure to L-DOPA	RatNHPHuman	Gardoni et al., 2006 [46]
Hallett et al., 2005 [47]
Mellone et al., 2015 [48]
Elevated NMDAR activation and glutamate release	RatNHPHuman	Nash et al., 2002 [49]
Calon et al., 2002 [44]
Ahmed et al., 2011 [50]
NMDAR antagonists reduce basal SPN hyperactivity and “unstable” responses to L-DOPA	RatNHP	Papa et al., 1995 [51]
Singh et al., 2018 [14]

AMPAR	Overexpression of receptors in the striatum	RatNHPHuman	Bagetta et al., 2012 [42]
Calon et al., 2002 [44]
Calon et al., 2003 [45]
AMPAR antagonists reduce basal SPN hyperactivity and “unstable” responses to L-DOPA	RatNHP	Fieblinger et al., 2014 [23]
Singh et al., 2018 [14]

Group ImGluRs	mGluR5 levels correlate with LID development and severity	RatNHPHuman	Crabbe et al., 2018 [52]
Morin et al., 2013; Morin et al., 2013 [53,54]
Ouattara et al., 2011 [55]
mGluR5 NAMs attenuate nigrostriatal degeneration	RatNHP	Armentero et al., 2006 [56]
Masilamoni et al., 2011 [57]

mGluR5 levels are elevated in LIDs	RatNHPHuman	Crabbe et al., 2018 [52]
Sanchez-Pernaute et al., 2008 [58]
Ouattara et al., 2011 [55]
mGluR5 binding and expression in the striatum increase following L-DOPA	RatNHPHuman	Crabbe et al., 2018 [52]
Samadi et al., 2008 [59]
Ouattara et al., 2011 [55]
mGluR5 NAMs reduce mGluR5 expression	RatNHP	Lea et al., 2006 [60]
Morin et al., 2013 [54]

Group IImGluRs	mGluR2/3 reduce excitatory drive on SPNs and LIDs	Rat	Pisani et al., 2000 [61]


Group IIImGluRs	mGluR4 reduces glutamate neurotransmission	RatNHP	Calabrese et al., 2022 [62]
Charvin et al., 2018 [62]

mGluR4 PAM decreases spontaneous glutamatergic transmission and restores bidirectional plasticity	Rat	Calabrese et al., 2022 [62]



## Data Availability

No new data were created or analyzed in this study. Data sharing is not applicable to this article.

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
