# Peer review of "Targeting Striatal Glutamate and Phosphodiesterases to Control L-DOPA-Induced Dyskinesia"

_cells, 2023, doi:10.3390/cells12232754_

Round 1

Reviewer 1 Report

Comments and Suggestions for Authors

cells-2722110

 The manuscript by Brik A. Kochoian and colleagues, entitled “Targeting Striatal Glutamate and Phosphodiesterases to Control L-DOPA-Induced Dyskinesia’’ aims to describe some therapeutic strategies that target Striatal Glutamate and Phosphodiesterases to control L-DOPA-induced dyskinesias (LID), in Parkinson's disease (PD).

Dysregulated glutamate signals in advanced stages of PD lead to hyperexcitability and hyperactivity of SPNs. This hyperactivity of SPNs has been reduced by treatment with NMDA receptor inhibitors resulting in improved LIDs. While amantadine, a drug that has an NMDAR antagonistic action, has been approved for the treatment of PD, there are other therapeutic targets with antidyskinetic action that are the topic of this review and are deeply described in the present work.

mGlu receptors (mGluR) have been extensively described as contributing to the pathogenesis of LID and, mGluR4 and mGluR5 receptors have received a lot of attention within this family. In fact, therapeutic agents for LIDs have been obtained by the antagonists of these two receptors.

Cyclic nucleotide levels appear to be involved in the induction of LIDs and they are important in the transduction of dopaminergic signals in SPNs. The enzymes that are responsible for degrading of these cyclic nucleotides are the phosphodiesterases (PDEs). The inhibition of PDE10A and other important PDEs, mostly expressed in SPNs, prevent dyskinesia in rats with PD.

Overall, the topic of the manuscript is very interesting as it offers a wide range of antiglutamatergic and PDE inhibition strategies useful for future development of novel LID therapies.

The Abstract is well written, with a good description of all the relevant data and the introduction and other paragraphs of the text are well described and presented.

However, I would like to bring attention to some minor inaccuracies that could help improve the manuscript.

Minor comments

English language and style are acceptable, although I have small changes to suggest as follows:

-(line 10), in the abstract " A large body of work during the past several decades HAVE been focused…" should read "A large body of work during the past several decades HAS been focused…";

-(line 93) It is not specified the acronym of NHPs;

-(line 115) standardize acronyms, e.g. iSPNs or i-SPNs;

-(line 153) "…activating adenylyl cyclase (AC) to SYNTHETIZE" should read "…activating adenylyl cyclase (AC) to SYNTHESIZE";

-(line 192) "…Striatal overexpression of NMDA and AMPA receptors HAVE been shown…" should read "… Striatal overexpression of NMDA and AMPA receptors HAS been shown…";

-(line 353) "…supporting the notion that the antidyskinetic properties of PDE10A inhibition IS driven by facilitation…" should read "… supporting the notion that the antidyskinetic properties of PDE10A inhibition ARE driven by facilitation…";

-(line 366) "…while genetic knockout in mice enhanced responsiveness to DA agonists and INCREASE locomotor activity. " should read "… while genetic knockout in mice enhanced responsiveness to DA agonists and INCREASED locomotor activity. ".

Major comments:

(lines 116-123) It needs to be rephrased the concepts and describe better the results obtained from the different studies cited: 1) Picconi et al. (citations 35 and 38) did not study synaptic plasticity identifying the two populations iSPN and dSPN; 2) Citations 36 and 37 described opposite synaptic plasticity results. Please clarify and amplify this concept describing all the data accordingly to the references.

Comments on the Quality of English Language

This is a nice review paper on an interesting and clinically relevant topic.

Reviewer 2 Report

Comments and Suggestions for Authors

The manuscript presented by Kochoian and coworkers is an interesting review about the glutamatergic dysregulation on L-DOPA-induced dyskinesias (LID). The review is focused on the various therapeutic options for managing dyskinesia targeting the different receptors of the glutamatergic system and the phosphodiesterases as regulators of levels of cycle nucleotides.

The manuscript is well done, well organized and correctly structured. The review covers the background of the field with relevant citations in the bibliography. Moreover, the figure is illustrative, and the tables are well-designed and explanatory. Nevertheless, some issues can be added or modified to improve the manuscript before publishing it:

1)      Abnormalities in the corticostriatal synapses and the glutamatergic system have a key role in the development of LID. However other neurotransmitter systems and factors, such as the abnormal release of dopamine by the serotonergic terminals (Prog Brain Res 2021:261:287-302. doi: 10.1016/bs.pbr.2021.01.032) or the neuroinflammation are also involved in the pathophysiology of LID (Br J Pharmacol 2020;177(24):5622-5641. doi: 10.1111/bph.15275). This concern should be mentioned in the introduction section.

3)      Since amantadine, as an NMDA antagonist, is the only FDA-approved treatment available for LID, it would be interesting to give more details about it, such as commenting on the history of the drug when it started to be applied, forms of amantadine currently available in the market or advantages and disadvantages of the treatment (J Neural Transm (Vienna). 2018;125(8):1237-1250. doi: 10.1007/s00702-018-1869-1)

4)      Interestingly, amantadine has also shown anti-inflammatory effects in in vitro and in vivo models of neuroinflammation and PD, reducing levels of TNF-α, IL-1β and NO. (Mol Neurobiol 2022 ;59(6):3703-3720. doi: 10.1007/s12035-022-02814-6; Neurobiol Aging 2012; 33(9):2145-59. doi: 10.1016/j.neurobiolaging.2011.08.011). Neuroinflammatory processes have been linked to dyskinesias. Therefore, amantadine could therefore exert its anti-dyskinetic properties by reducing neuroinflammation. Additionally, a very recent review deals with the relationship between N-methyl-D-aspartic acid receptor and neuroinflammation in LID (Front Immunol 2023:14:1253273. doi: 10.3389/fimmu.2023.1253273). This issue could be also mentioned.

5)      The section “2.2. Dopamine signaling” cited the relation between de adenosine A2A receptors and the D2 dopamine receptors: “Notably, adenosine 2A receptors (A2AR) in iSPNs counteract D2Rs activating AC, increasing cAMP synthesis, likely for preventing excessively low levels cAMP”. In relation with this is important to mention that the A2A and the D2 receptors are formed heteromers in the striatal neurons. Moreover, a recently published study showed the interaction of a a high proportion of dopamine D2 receptors with adenosine A2A receptors in experimental models of Parkinson's disease and dyskinesia, suggesting benefits of antiparkinsonian treatment with adenosine receptor blockers. (Neurobiol Dis. 2023 Oct 31:188:106341. doi: 10.1016/j.nbd.2023.106341)

6)      In Figure 1, symbols for D1R and mGluR1/5 are quite similar, please change the symbol/colour or add the name of each receptor with small letters in the scheme.   

7)      Tables are well designed; however, it would be more graphic to include in addition to the reference number, the author and/or the journal name.
